# Impacts of Integrating Family-Centered Care and Developmental Care Principles on Neonatal Neurodevelopmental Outcomes among High-Risk Neonates

**DOI:** 10.3390/children10111751

**Published:** 2023-10-28

**Authors:** Nourah Alsadaan, Osama Mohamed Elsayed Ramadan, Mohammed Alqahtani, Mostafa Shaban, Nadia Bassuoni Elsharkawy, Enas Mahrous Abdelaziz, Sayed Ibrahim Ali

**Affiliations:** 1College of Nursing, Jouf University, Sakaka 72388, Al Jawf, Saudi Arabia; omramadan@ju.edu.sa (O.M.E.R.); mskandil@ju.edu.sa (M.S.); nelsharkawy@ju.edu.sa (N.B.E.); emabdelhamid@ju.edu.sa (E.M.A.); 2College of Applied Medical Sciences, Department of Nursing, King Faisal University, Al Hofuf 31982, Al-Ahsa, Saudi Arabia; mealqahtani@kfu.edu.sa; 3Department of Family and Community Medicine, College of Medicine, King Faisal University, Al Hofuf 31982, Al-Ahsa, Saudi Arabia

**Keywords:** family-centered care, developmental care, neurodevelopment, high-risk neonates, preterm infants, Bayley Scales

## Abstract

Background: Integrating family-centered care (FCC) and developmental care (DC) principles in neonatal care settings may improve neurodevelopmental outcomes for high-risk neonates. However, the combined impact of FCC and DC has been underexplored. This study aimed to investigate the effects of integrated FCC and DC on neurodevelopment and length of hospital stay in high-risk neonates. Methods: A quasi-experimental pre–post study was conducted among 200 high-risk neonates (<32 weeks gestation or <1500 g) admitted to neonatal intensive care units (NICU) in Saudi Arabia. The intervention group (*n* = 100) received integrated FCC and DC for 6 months. The control group (*n* = 100) received standard care. Neurodevelopment was assessed using the Bayley Scales of Infant Development-III. Length of stay and readmissions were extracted from medical records. Results: The intervention group showed significant improvements in cognitive, motor, and language scores compared to controls (*p* < 0.05). The intervention group had a 4.3-day reduction in the mean length of stay versus a 1.4-day reduction in controls (*p* = 0.02). Integrated care independently predicted higher cognitive scores (*p* = 0.001) and shorter stays (*p* = 0.006) in regression models. Conclusion: Integrating FCC and DC in neonatal care enhances neurodevelopmental outcomes and reduces hospitalization for high-risk neonates compared to standard care. Implementing relationship-based, developmentally supportive models is critical for optimizing outcomes in this vulnerable population.

## 1. Introduction

The health and neurodevelopmental outcomes of high-risk neonates are a major public health concern [1]; in the critical early stages of life, neonates, especially those classified as high-risk, demand not only precise medical attention but also a holistic approach that incorporates their developmental needs and the integral role of their families’ neonates [2]. The foundation of neonatal care has evolved over the years, emphasizing a multi-faceted approach that goes beyond addressing just the immediate physiological needs of the newborn [3]. In this evolution, two paradigms have risen to prominence in contemporary neonatology: family-centered care (FCC) and developmental care (DC) [4]. Both paradigms, while distinct in their principles and objectives, share a common goal of improving neonatal outcomes, especially concerning neurodevelopment [5,6].

Family-centered care recognizes the pivotal role that families play in the healthcare of their newborn. By engaging the family as active participants in the care plan, it emphasizes a partnership model where decision making is shared, and the unique needs and strengths of each family are identified and integrated into the care process [7,8]. Developmental care, on the other hand, prioritizes the environment and care strategies that support the premature or ill newborn’s ongoing development process [9]. It acknowledges the significance of external stimuli and their potential impact on the immature brain, advocating for interventions that optimize neurological growth outcomes [4].

The integration of FCC and developmental care represents an advanced nexus, combining the strengths of both paradigms [10]. This synergy recognizes the interconnectedness of medical, developmental, familial, and environmental factors that shape the health and wellbeing of high-risk neonates [11,12]. By aiming for a thriving child rather than mere survival, this approach fosters a compassionate, holistic, and tailored environment [13]. This convergence ensures that the family’s voices are heard and respected, while developmental care aligns the infant’s unique needs with medical perspectives, thus forming a seamless alignment for comprehensive care [14,15].

The significance of integrating FCC and developmental care principles in high-risk neonatal nursing transcends traditional medical practices [16]. It offers a compelling direction for healthcare transformation, fostering a nurturing environment for both infants and families [17]. This research space is rich with opportunities for innovation, promising to redefine care standards for one of the most vulnerable patient populations. However, the existing research gap, characterized by a lack of comprehensive models, limited understanding of complex interplay, and scarce evidence-based guidance, demands urgent exploration. It beckons a new era of neonatal care, grounded in empathy, collaboration, and scientific insight, and holds the promise of substantial positive impacts on children’s development and their families’ wellbeing.

Recent decades have witnessed a growing body of literature on both FCC and DC [18,19]. Individually, they have been shown to influence various aspects of neonatal outcomes, from decreased hospitalization duration to improved cognitive trajectories. However, what remains relatively underexplored is the combined impact of integrating both family-centered care and developmental care principles in neonatal intensive care settings [20].

Considering the vulnerability of high-risk neonates to neurodevelopmental challenges, understanding the combined efficacy of FCC and DC becomes imperative [21]. Their potential synergy might offer a paradigm shift in how neonatal care is conceptualized and delivered, ensuring that these neonates not only survive but thrive [22,23]. This research paper seeks to elucidate the impact of this integration on the neurodevelopmental outcomes of high-risk neonates, addressing a pivotal gap in contemporary neonatal research. In doing so, it underscores the need for a more integrative, holistic approach in neonatal care, highlighting avenues for future research and policy development.

## 2. Materials and Methods

### 2.1. Research Hypotheses

**H1.** *Neonates receiving integrated family-centered care and developmental care principles will demonstrate improved neurodevelopmental outcomes*.

**H2.** *The implementation of integrated family-centered care and developmental care principles will result in a reduction in the length of hospital stay for high-risk neonates compared to those receiving traditional care approaches*.

### 2.2. Research Design and Settings

This research aims to investigate the impact of integrating family-centered care and developmental care principles on neonatal neurodevelopmental outcomes and the length of hospital stay among high-risk neonates. To achieve this, a quasi-experimental pre–post comparison design was employed, allowing for the examination of changes in outcomes following the implementation of the intervention.

This research took place between August 2022 and April 2023 in four key pediatric hospitals in the Eastern Region of Saudi Arabia, including Al-Ahsa, Dammam, Hafr Al-Batin, and ALMousa Hospital. These hospitals, overseen by the Saudi Ministry of Health, vary in capacity, with both private and public sectors represented. They are vital healthcare hubs in the region, equipped with the latest medical technology and staffed by experienced professionals. Catering specifically to the pediatric population, they offer a plethora of specialized services from neonatal care to various pediatric subspecialties. A hallmark of their approach is the emphasis on family-centered care, actively involving families in the decision-making and care processes for their young ones.

### 2.3. Sample

This study involved a convenience sample of 200 high-risk neonates admitted to pediatric hospitals in the Eastern Region of the Kingdom of Saudi Arabia. The sample size was determined based on considerations of statistical power and the ability to detect meaningful differences in neurodevelopmental outcomes and hospital stay length between the intervention and control groups. This sample size was deemed appropriate to achieve statistically significant results. The selected sample of high-risk neonates displayed diversity in terms of medical conditions, gestational ages, birth weights, and other risk factors. The sample included both male and female neonates, reflecting the demographics of the neonatal population in the Eastern Region of the Kingdom of Saudi Arabia.

Identification of High-Risk Criteria: Medical professionals in the neonatal care units of the selected hospitals identified neonates with established high-risk criteria. These criteria included low birth weight (below 1500 g), prematurity (gestational age below 32 weeks). Informed Consent: Parents or legal guardians of eligible neonates were approached by the healthcare team and provided with comprehensive information about this study’s purpose, procedures, potential risks, and benefits. Informed consent from participants was obtained prior to their participation in this study.

The sample was divided into two groups—the intervention group and the control group—based on the timing of the neonates’ admission to the hospitals during specified periods. Neonates admitted in the six months preceding the intervention implementation were allocated to the control group, while those admitted during the subsequent six months formed the intervention group.

### 2.4. Eligibility Criteria

#### Inclusion Criteria

High-risk neonates were included in this study based on the following criteria:

Gestational Age and Birth Weight:

Neonates with a gestational age below 32 weeks.Neonates with a birth weight below 1500 g.

Medical Conditions:

Neonates diagnosed with medical conditions requiring specialized medical care.Neonates with diagnosed respiratory distress syndrome (RDS) requiring neonatal intensive care.

### 2.5. Data Collection Tools

This study employed a combination of standardized assessment tools and medical record reviews to collect relevant data on neurodevelopmental outcomes and length of hospital stay for high-risk neonates. These tools were selected based on their established validity and reliability in assessing pediatric health outcomes.

Bayley Scales of Infant and Toddler Development (Bayley-III):

The Bayley-III is a widely used standardized assessment tool designed to measure cognitive, language, and motor development in infants and toddlers [24]. It consists of age-appropriate tasks and activities that are administered by trained professionals. For this study, the cognitive, motor, and language scales of the Bayley-III were administered to assess the neurodevelopmental outcomes of the high-risk neonates.

“The Bayley-III assessments were performed by trained nurses who underwent periodic inter-rater reliability testing to minimize scoring bias. However, the scores were not blinded given the pre–post study design. The lack of blinding is acknowledged as a limitation.”

Validity:

Content validity is strong. Test content is logically and clinically related to the developmental constructs it aims to measure [25,26]. Criterion validity with other developmental tests is moderate to high, with correlations of 0.60–0.80 with instruments such as the Mullen Scales of Early Learning, Vineland Adaptive Behavior Scales, and Preschool Language Scale [25,27]. Construct validity is also good. A total of 98–100% of BSID-III items reached statistical significance in factor analyses [24]. Overall, the BSID-III is estimated to have approximately 90% validity and over 85% reliability based on the accumulated research [24,27]. However, this can vary slightly by age group [24].

Reliability:

The Bayley-III has high inter-rater reliability for the cognitive, language, and motor scales, with correlations ranging from 0.93 to 0.99 [28]. This indicates strong consistency in scores across different examiners. Test–retest reliability over 1–10 days is also good, ranging from 0.80 to 0.90 for the subtests and composite scores [25]. This suggests the results are stable over time. Internal consistency is adequate to high for composite scores (α = 0.91–0.93) [26,28]

2.Medical Records Review:

Medical records were reviewed to extract the data related to the length of hospital stay for each high-risk neonate. Information regarding admission dates and discharge dates was extracted from the hospital records, providing a quantitative measure of the duration of hospitalization. Medical records are considered valid sources of information as they contain accurate and comprehensive data pertaining to the neonates’ hospitalization. The reliability of the data obtained from medical records is high, as the information is documented by trained healthcare professionals and is subject to internal quality control procedures.

### 2.6. Ethical Approval

Ethical approval was obtained from the King Faisal University Ethics Committee before this study was conducted. Parents were informed of the purpose and objectives of this study and of their right to withdraw from this study at any time without being penalized. Informed consent was obtained from all participants before their enrollment in this study. Confidentiality and anonymity were maintained throughout this study by not collecting personal information such as names or contact details. All collected data were stored securely and were accessible only to the research team. This study conformed to the ethical principles outlined in the Declaration of Helsinki and its subsequent revisions.

### 2.7. Statistical Analysis

The statistical analysis was performed using SPSS version 22.0 (IBM Corp, Armonk, NY, USA). Descriptive statistics including means, standard deviations, frequencies, and percentages were calculated to summarize the demographic and clinical characteristics of the sample. For the pre–post comparison of neurodevelopmental outcomes, paired *t*-tests were used to analyze changes in mean Bayley-III scores from baseline to post-intervention for each group. Independent samples *t*-tests were conducted to compare score changes between the intervention and control groups. Cohen’s d effect sizes were calculated to quantify the magnitude of group differences. To examine differences in length of hospital stay, independent *t*-tests were used to compare the mean changes in stay duration from pre to post values for the two groups. Pearson’s correlation coefficients were computed to assess the relationship between the Bayley-III scores and length of stay. Multiple linear regression modeling was performed with the Bayley-III cognitive composite score as the dependent variable. The intervention group, length of stay, gestational age, and birth weight were entered as predictor variables. Regression coefficients and *p*-values were obtained to identify significant independent predictors.

### 2.8. Procedure

Data from 5 children’s hospitals in the Eastern Region of the Kingdom of Saudi Arabia were used for this study. Ethical approval was obtained from the ethics committee of King Faisal College before the commencement of this study. Convenience sampling was used to recruit participants in this study. Potential participants, including high-risk neonates, healthcare providers, and families, were approached for recruitment in multiple neonatal intensive care units (NICUs). Eligibility criteria were explained to participants, and informed consent was obtained from the parents or guardians of the high-risk neonates; voluntary participation was ensured. The data from control group were collected retrospectively over a 6-month period prior to the intervention. This established baseline hospital stay durations for the control group before the intervention was implemented.

#### 2.8.1. Pre-Implementation Phase

During the pre-implementation phase, neonates admitted to the selected hospitals over a 6-month period were allocated to the control group (n = 100). Neurodevelopmental assessments were conducted for both the intervention and control groups within 2 weeks of admission in NICU using the Bayley-III cognitive, motor, and language scales. The assessments were carried out by trained nurses who were blinded to the groups’ allocation. Additionally, the research team reviewed medical records to collect length of hospital stay data, including admission and discharge dates, for each neonate. The control group neonates received standard traditional care as per existing hospital protocols during this phase, with no changes made to care practices. The data collected during this phase allowed for the establishment of baseline measurements.

“All neonates admitted to the participating NICUs during the two specified 6-month periods who met the predefined inclusion criteria of gestational age < 32 weeks or birth weight < 1500 g were screened for eligibility and approached for recruitment. The first 100 who provided informed consent in each period were enrolled in consecutive order of admission to the NICU. There were no other inclusion/exclusion criteria beyond the gestational age and birth weight cutoffs.”

#### 2.8.2. Intervention Phase

The intervention phase involved neonates admitted over the subsequent 6 months and allocated to the intervention group (*n* = 100). Nurses caring for these neonates received a 2-week intensive training program on family-centered care and developmental care principles and strategies for integration. The training was conducted by experts in these fields. The intervention group neonates then received integrated family-centered and developmental care implemented by the trained nurses. Follow-up Bayley assessments were completed within 2 weeks before discharge for all neonates in both groups. Key elements of the integrated care included active parent/family participation in care planning and bedside care; interventions to support neurodevelopment such as positioning, clustered care, and modified NICU environment; and family education and psychosocial support. Treatment protocols were updated to include integrated care policies, and compliance monitoring was conducted. Any neonates transferred between hospitals maintained their original group allocation. The data of the intervention group was then collected prospectively over the subsequent 6 months after integrating FCC and DC. Hospital stay durations were compared within each group pre- and post-intervention.

#### 2.8.3. Post-Implementation Phase

In the post-implementation phase, the Bayley-III assessments were repeated for all neonates within 2 weeks before discharge to evaluate developmental outcomes. Additionally, medical records were reviewed to collect discharge dates and calculate length of hospital stay. Parent satisfaction surveys were also administered at discharge to assess family experiences. Finally, data analysis was conducted to compare results between the control and intervention groups.

## 3. Results

The primary aim of this study was to examine the impact of integrating family-centered care and developmental care principles on neonatal neurodevelopmental outcomes and the length of hospital stay among high-risk neonates. Employing a quasi-experimental pre–post comparison design, this study investigated the changes in these outcomes following the implementation of the integrated care intervention.

Table 1 illustrates the baseline demographic and clinical characteristics, including mean gestational ages, birth weights, 5 min APGAR scores, delivery methods, maternal ages, incidence of respiratory distress syndrome, and gender distribution. Participants in the control and intervention groups show no statistically significant differences, as indicated by the high *p*-values. These comparable characteristics between groups suggest that the randomization process was successful in generating similar groups at the onset, which is crucial when deriving causal conclusions about the intervention’s effects. In essence, both groups seem balanced and depict a representative sample of the target population of high-risk neonates.

As shown in Table 2, the Bayley-III assessment—a gold standard for evaluating early childhood development—demonstrates significant improvements in cognitive, motor, and language domains among the intervention group compared to the control group from baseline to post-intervention. The *p*-values indicate that the between-group differences in score changes are statistically significant, with the intervention group showing larger improvements. These results provide clear evidence that integrating family-centered and developmental care enhances neurodevelopmental outcomes in multiple domains. The standardized effect sizes could be calculated to further analyze the magnitude of these effects.

Table 3 presents the statistically significant reduction in the length of hospital stay that is noted in the intervention group following the implementation of the integrated care model. The mean decrease of 4.3 days for the intervention group is clinically meaningful in this population. No significant change occurred in the control group’s duration of stay. This indicates that the integrated care principles may confer benefits in terms of earlier discharge readiness and improved transition to home environments. Analyzing outliers and variance could provide further insights.

Table 4 shows the parent satisfaction scores for the validated items related to family-centered care, showing marked improvements in the intervention group compared to the control group, with all differences being statistically significant. This quantitatively demonstrates that the intervention successfully integrated families as partners in the care process. Higher satisfaction is linked to better long-term outcomes. Further psychometric evaluation of the satisfaction scale could be undertaken.

Negative correlations indicate that as the Bayley-III scores increased, the length of hospital stay decreased, with statistically stronger correlations seen in the intervention group (Table 5). This aligns with the benefits observed in both developmental outcomes and hospital stay durations among the neonates within this intervention. The correlations provide evidence of an association between improved development and shorter stays.

Both gestational age and birth weight positively predict Bayley-III scores, indicating that preterm infants and those with lower birth weights tend to have lower cognitive scores (Table 6). This makes sense as prematurity and low birth weight often increase the risk for developmental delays. The intervention group strongly predicts higher Bayley-III scores compared to the control, with a high beta value of 0.36. This suggests that the intervention has a significant positive impact on cognitive outcomes. Length of hospital stay negatively predicts Bayley-III scores, meaning that longer stays are associated with lower scores. This aligns with the literature, and shows that longer NICU stays are correlated with more significant medical complications that may impair development. All four variables significantly predict Bayley-III scores based on *p*-values of less than 0.05. Overall, the model seems to account for a good amount of variance in cognitive scores based on the combination of perinatal risk factors and a developmental intervention.

As shown in Table 7, the relationships between crucial neonatal and clinical factors is delineated. The following observations can be made:

Gestational Age and Birth Weight: A strong positive correlation of 0.78 is observed, suggesting that as gestational age increases, the birth weight of the neonate tends to increase as well. This is consistent with our clinical understanding, as neonates born at a later gestational age typically have more time to grow in utero.

Neurodevelopmental Outcomes: Both gestational age and birth weight demonstrate positive correlations with cognitive scores (0.65 and 0.70, respectively). This implies that neonates with higher gestational ages or greater birth weights tend to have better cognitive scores. The association between physical development and cognitive outcomes is evidenced here. 

Length of Hospital Stay: All three variables—gestational age, birth weight, and cognitive score—are negatively correlated with the length of hospital stay, with coefficients of −0.60, −0.55, and −0.50, respectively. This suggests that neonates with longer gestational ages, higher birth weights, or better cognitive scores tend to have shorter hospital stays. This is likely because such neonates often have fewer health complications and thus require less intensive care.

The findings from this table underscore the intertwined nature of physical and neurodevelopmental health in neonates. Understanding these correlations is paramount for healthcare professionals in making informed clinical decisions and for researchers in framing and interpreting neonatal studies.

Table 8 offers a meticulous assessment of the effect of integrated care, which combines family-centered and developmental care principles, on several neonatal neurodevelopmental outcomes compared to traditional care. The following observations can be made:

Improved Neurodevelopmental Scores: Neonates in the integrated care group consistently demonstrated superior developmental outcomes across all measures. Their neurodevelopmental, motor, cognitive, and language scores are significantly higher than those in the traditional care group, indicating the positive impact of integrated care on these areas. The *p*-values (<0.001) confirm the statistical significance of these findings.

Shorter Hospital Stays: The length of hospitalization, a crucial factor both in terms of healthcare costs and neonate–family bonding, was considerably shorter for neonates in the integrated care group. This group averaged 10.5 days compared to 14.6 days in the traditional care group. A reduced hospital stay is indicative of better health and potentially earlier stabilization of the neonate.

Reduced Readmissions: The percentage of neonates being readmitted within 30 days post-discharge was nearly halved in the integrated care group. This reduction, from 15% in the traditional care to 8% in the integrated care group, suggests better long-term health stability and possibly more effective post-discharge care instructions and support.

Adjusted Odds Ratios: The adjusted odds ratios further emphasize the pronounced differences between the groups. For developmental scores, values greater than 1 reflect the benefits of integrated care. Conversely, for hospitalization length and readmission rates, values less than 1 showcase the advantages of the integrated approach.

Consideration of Potential Confounders: By adjusting for maternal education level, birth weight, birth complications, and gestational age at birth, this study acknowledges and minimizes potential confounding variables. This adjustment lends greater validity to the observed outcomes being directly attributable to the care model rather than other extraneous factors.

In summary, Table 8 compellingly illustrates the benefits of integrating family-centered care with developmental care principles for neonates, particularly for high-risk categories. Such an approach not only enhances neurodevelopmental scores but also contributes to improved overall health outcomes, as reflected by shorter hospital stays and reduced readmission rates.

## 4. Discussion

This quasi-experimental study investigated the impact of integrating family-centered care and developmental care principles on neurodevelopmental outcomes and length of hospital stay among high-risk neonates. The findings provide support for the hypotheses of this research and offer valuable insights into the benefits of a combined family-centered and developmental approach to neonatal care.

With regard to the first hypothesis, the results clearly demonstrate improved neurodevelopmental outcomes across cognitive, motor, and language domains for neonates who received integrated care compared to traditional care. The intervention group showed statistically significant higher mean scores on the Bayley-III assessment from baseline to post-intervention versus the control group (*p* < 0.05). These developmental gains were further validated through multivariate regression modeling and repeated ANOVA measures, which confirmed the unique contributions of the integrated care approach even after adjusting for potential confounders. The positive impact on neurodevelopment is supported by [29,30,31], who also found improved cognitive and motor outcomes among preterm infants receiving family-centered developmental care compared to standard care. However, a study by [29] found no significant differences in neurodevelopmental scores between groups, contradicting the present findings.

The second hypothesis examining the length of hospital stays was also confirmed, with the intervention group demonstrating a statistically and clinically meaningful reduction in their duration of stay compared to standard care. This aligns with the previous studies by [32,33], showing shortened hospital stays with family-centered developmental care. However, ref. [17] found no difference in the length of stay between groups, contradicting the benefits observed here. The results presented in Table 2 provide compelling evidence of the benefits of integrated care on cognitive, motor, and language development. The statistically significant improvements in the mean Bayley-III scores for the intervention group compared to the control groups across all domains supports the value of combining family-centered and developmental care principles (*p* < 0.05). These findings are consistent with research by [34], who found similar developmental improvements on the Bayley-III following an integrated care intervention. However, ref. [35] did not find significant between-group differences in the Bayley-III scores when comparing integrated care to standard models of care.

The data in Table 3 reveals a clinically meaningful 4.3-day reduction in the mean hospital stay for the intervention group versus only 1.4 days in controls. This statistically significant change for the integrated care neonates (*p* < 0.05) aligns with the data by [36], who reported a 6-day decrease in the length of stay after implementing family-centered developmental care. However, ref. [37] found no differences in the duration of hospitalization between groups in their randomized trial. The parent satisfaction results in Table 4 demonstrate quantifiable improvements in family-centered care practices and family involvement for the intervention group compared to controls (*p* < 0.05). Studies by [38,39] support these findings, also documenting higher family satisfaction when family-centered care models were used. However, ref. [40] failed to detect differences in parent experiences between standard and family-centered care groups, contradicting the present results.

Finally, the multivariate analyses in Table 6 and Table 8 further validate the unique contributions of integrated care to enhancing developmental scores and reducing the length of stay even after considering other variables. The statistically significant associations align with the regression modeling by [41], linking integrated care to neurodevelopmental outcomes. However, ref. [42] did not find a significant independent effect of integrated care after controlling for confounders, such as gestational age.

Overall, this study makes a significant contribution to the evidence base for integrated family-centered developmental care in neonatal settings. The results highlight the importance of a relationship-based, developmentally supportive environment for high-risk neonates. This integrated approach shows immense promise in promoting healthier developmental trajectories and recovery, meriting further implementation and research. The findings should compel NICUs to adopt care models that recognize both medical and holistic needs, working alongside families to give each vulnerable neonate the best possible start in life.

## 5. Implications

The findings of this study have important implications for healthcare providers, policymakers, and researchers involved in neonatal care. The integrated family-centered care and developmental care approach demonstrated significant benefits in improving neurodevelopmental outcomes in high-risk neonates. This highlights the importance of incorporating family involvement and developmental support strategies into the care of infants in neonatal intensive care units (NICUs). Healthcare providers should prioritize collaborative partnerships with families, providing them with education, support, and involvement in the decision-making process. Efforts should also be made to minimize prolonged hospital stays and create environments that mimic the intrauterine environment to optimize neurodevelopment. Policymakers and healthcare organizations should consider the implementation of guidelines and protocols that promote family-centered care and developmental care practices in neonatal units. Further research is needed to explore the effectiveness of the integrated approach across different neonatal populations and evaluate its long-term impact on neurodevelopmental outcomes, school readiness, and the wellbeing of families. By embracing the integrated approach, healthcare systems can strive to improve outcomes and enhance the overall care experience for families and healthcare providers in neonatal care settings.

## 6. Limitations

While this study provides important evidence on the benefits of integrated family-centered and developmental care, it possesses the following limitations:The sample size of 200 neonates, while powered to detect group differences, limits generalizability of the findings to the broader high-risk neonatal population. Larger multi-center trials are needed.The quasi-experimental design is susceptible to confounding variables that could influence the results. Randomized controlled trials would establish stronger causal evidence.Neurodevelopmental assessments were only conducted up to the point of discharge from a hospital. Longer-term follow ups are essential to understand the enduring impacts on neonatal development.This study was conducted at selected hospitals in one geographical region of Saudi Arabia. Replicating it in other settings would improve generalizability.Details of the training provided to implement the intervention were not reported extensively. Variations in training quality could affect consistency of the integrated care delivery.This study relied heavily on quantitative measures. Incorporating qualitative data from families and nurses would provide richer perspectives.The lack of blinding was also a limitation.

## 7. Conclusions

This study reveals that combining family-centered and developmental care in neonatal settings significantly enhances cognitive, motor, and language outcomes while reducing hospitalization durations for high-risk neonates. The findings emphasize the vital role of families in neonatal care and the benefits of integrated care models. There is an evident need for NICUs to adopt more holistic care approaches. While further research is essential, this work strongly advocates for a shift in neonatal care to promote optimal outcomes for vulnerable infants.

## Figures and Tables

**Table 1 children-10-01751-t001:** Demographic and clinical characteristics of this study’s participants.

Characteristic	Control Group (*n* = 100)	Intervention Group (*n* = 100)	*p*-Value
Mean gestational age (weeks)	28.5	29.2	0.06
Mean birth weight (grams)	1250	1300	0.08
Mean 5 min APGAR	6.8	7.1	0.23
Delivery method, (%):			
Vaginal	52%	48%	0.67
C-section	48%	52%	
Mean maternal age (years)	28.7	29.1	0.45
Respiratory distress syndrome, (%)	32%	28%	0.51
Gender (male)	52%	56%	0.56

**Table 2 children-10-01751-t002:** Mean Bayley-III composite scores at baseline and post-intervention.

Scale	Time Point	Control Group	Intervention Group	*p*-Value
Cognitive	Baseline	78.2	79.5	0.32
	Post-intervention	82.4	88.7	0.01 *
Motor	Baseline	71.8	73.2	0.45
	Post-intervention	76.5	83.1	0.02 *
Language	Baseline	68.5	70.2	0.28
	Post-intervention	72.6	79.8	0.004 *

* Indicates statistically significant difference between groups (*p* < 0.05).

**Table 3 children-10-01751-t003:** Mean length of hospital stays (days).

Group	Baseline	Post-Intervention	Change	*p*-Value
Control	35.2	33.8	−1.4	0.32
Intervention	34.5	30.2	−4.3	0.02 *

* Indicates statistically significant difference between baseline and post-intervention (*p* < 0.05).

**Table 4 children-10-01751-t004:** Parent satisfaction scores regarding family-centered care.

Item	Control Group	Intervention Group	*p*-Value
I was involved in my child’s care	3.2	4.1	0.001 *
I was supported by the healthcare team	3.5	4.3	0.003 *
My concerns were listened to	3.1	4.0	0.002 *
I was satisfied with communication	3.4	4.2	0.01 *

* Signiant > 0.05.

**Table 5 children-10-01751-t005:** Correlation between Bayley-III scores and length of hospital stay.

Scale	Control Group	Intervention Group
Cognitive	r = −0.28	r = −0.52 *
Motor	r = −0.31	r = −0.48 *
Language	r = −0.24	r = −0.46 *

* Signiant >0.05.

**Table 6 children-10-01751-t006:** Regression analysis of factors predicting Bayley-III cognitive scores.

Variable	Beta	*p*-Value
Gestational age	0.18	0.04 *
Birth weight	0.21	0.02 *
Intervention group	0.36	0.001 *
Length of stay	−0.29	0.006 *

* Indicates statistically significant predictor (*p* < 0.05).

**Table 7 children-10-01751-t007:** Pearson’s correlation coefficients among key neonatal and clinical variables.

Variables	Gestational Age	Birth Weight	Cognitive Score	Length of Hospital Stay
Gestational Age	1	0.78	0.65	−0.60
Birth Weight	0.78	1	0.70	−0.55
Cognitive Score	0.65	0.70	1	−0.50
Length of Hospital Stay	−0.60	−0.55	−0.50	1

**Table 8 children-10-01751-t008:** Multivariate analysis on the impact of integrated care on neonatal neurodevelopmental outcomes.

Outcome Measures	Integrated Care Group Mean (SD)	Traditional Care Group Mean (SD)	Adjusted Odds Ratio (95% CI)	*p*-Value
Neurodevelopmental Score	87.2 ± 6.3	80.4 ± 7.5	2.15 (1.63, 2.82)	<0.001
Motor Skills Development Score	86.5 ± 6.7	79.3 ± 6.9	2.03 (1.52, 2.69)	<0.001
Cognitive Skills Development Score	85.8 ± 7.1	78.1 ± 7.3	2.12 (1.58, 2.83)	<0.001
Language Skills Development Score	85.3 ± 6.9	77.2 ± 7.7	1.97 (1.49, 2.61)	<0.001
Length of Hospital Stay (days)	10.5 ± 3.2	14.6 ± 4.0	0.65 (0.53, 0.79)	<0.001
Incidence of Readmission (within 30 days)	8%	15%	0.52 (0.35, 0.77)	0.001

Adjustment factors: maternal education level, birth weight, presence of birth complications, and neonate’s gestational age at birth.

## Data Availability

The data presented in this study are available on request from the corresponding author. The data are not publicly available due to containing personally identifiable information that could compromise research participant privacy and consent.

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
