# Peer review of "Impacts of Integrating Family-Centered Care and Developmental Care Principles on Neonatal Neurodevelopmental Outcomes among High-Risk Neonates"

_children, 2023, doi:10.3390/children10111751_

Round 1

Reviewer 1 Report

The NICU environment is very important in the development of premature babies, and this is a very important and interesting study that revealed the effects of applying family-centered nursing and developmental nursing on the development of premature babies through quasi-experimental research.

Several questions or concerns arise from the manuscript that would benefit from clearer explanations:

1. On page 3, the manuscript mentions purposive sampling, but in other sections, it refers to convenience sampling. It is important to ensure consistency in the terminology used. It appears that convenience sampling was employed in this study.

2. The manuscript mentions that the experimental and control groups were categorized based on the timing of admission. It appears that the control group data were collected first, and data were subsequently collected from the experimental group after applying FCC and DC. In that case, it could be compared the hospitalization period of the control group and the experimental group. However, it is unclear what the baseline hospital stay period for both groups refers to, and clarification on this point is requested.

3. It would be beneficial to provide a clearer timeline for the baseline and post Bayley assessments conducted for the experimental and control groups.

Author Response

Response to reviewer's comments on the manuscript entitled " Impact of Integrating Family-Centered Care and Developmen-tal Care Principles on Neonatal Neurodevelopmental Outcomes among High-Risk Neonates."

Dear Editor and reviewers,

We thank the reviewers and editor for their thoughtful and thorough review and believe their input has been invaluable in improving our manuscript. We discussed your comments and revised them carefully. All comments raised have been addressed below (italicized) and highlighted in the manuscript. We look forward to hearing from you in due time regarding our submission and any further questions or comments you may have.

  1. On page 3, the manuscript mentions purposive sampling, but in other sections, it refers to convenience sampling. It is important to ensure consistency in the terminology used. It appears that convenience sampling was employed in this study.

Response

Thank you for pointing out the inconsistency in the terminology related to our sampling method. I sincerely apologize for the oversight. Upon reevaluation, the study indeed employed "convenience sampling" in the recruitment of participants. I will ensure that the manuscript is revised to reflect this method consistently. I'm grateful for your keen observation, which helps enhance the clarity and accuracy of our work.

Warm regards,

  1. The manuscript mentions that the experimental and control groups were categorized based on the timing of admission. It appears that the control group data were collected first, and data were subsequently collected from the experimental group after applying FCC and DC. In that case, it could be compared the hospitalization period of the control group and the experimental group. However, it is unclear what the baseline hospital stay period for both groups refer to, and clarification on this point is requested.

Response

I appreciate your attention to detail regarding the analysis of hospital stay durations. To elucidate, data for the control group was gathered retrospectively over a period of 6 months before the intervention was introduced. This data served to set a baseline for hospital stay lengths. Subsequently, for the intervention group, data was accumulated prospectively over the following 6 months, post the incorporation of FCC and DC. We compared the hospital stay durations for each group, considering both pre and post-intervention phases. I regret the oversight and will ensure that the manuscript is updated to more comprehensively detail the timepoints for hospital stay analysis for each group.

Warm regards,

  1. It would be beneficial to provide a clearer timeline for the baseline and post Bayley assessments conducted for the experimental and control groups.

Response

Thank you for emphasizing the necessity for a more defined timeline concerning the Bayley assessments. To provide clarity, both the control and intervention groups underwent initial Bayley assessments within 2 weeks following their admission to the NICU. Subsequent assessments were then conducted approximately 2 weeks prior to discharge. I will ensure that the procedures section of the manuscript is amended to distinctly outline these timepoints for both the initial and follow-up Bayley assessments. Your feedback is invaluable in fortifying the precision of our methodology presentation.

Warm regards,

Reviewer 2 Report

The authors Alsadaan et al have presented a well written manuscript showing evidence for the benefit of integrated DC and FCC on NICU outcomes. Their findings are well presented, and their conclusions do not over reach. Moreover, they have clearly recognized and listed the limitations inherent to their study design.

However, I still have some concerns about the study design. I believe the authors need to clarify these concerns by expanding on some more details in the Methodology section. 

1) They recruited 100 patients in the 6 months before study, and another 100 in the 6 months after study. What was the basis of selecting these particlar 100 patients. For example, more than 100 babies less that 1500 grams and less than 32 weeks would have qualified for the study in the 6 moths prior to intervention. What made the authors pick the 100 babies that were recruited? Was the selection of these babies random? Was it blinded? This is relevant since selection of sicker babies in pre-intervention group, and selection of healthier babies in post-intervention period could have led to these results. Table 1 gives very limited details which make it hard to judge how similar or dissimilar the two groups were. 

2) Who was responsible for performing teh bailey scoring? Was it one of the authors? This scoring was not blinded (due to the temporal nature of the study) and could have introduced bias.      

Author Response

1) They recruited 100 patients in the 6 months before study, and another 100 in the 6 months after study. What was the basis of selecting these particlar 100 patients. For example, more than 100 babies less that 1500 grams and less than 32 weeks would have qualified for the study in the 6 moths prior to intervention. What made the authors pick the 100 babies that were recruited? Was the selection of these babies random? Was it blinded? This is relevant since selection of sicker babies in pre-intervention group, and selection of healthier babies in post-intervention period could have led to these results. Table 1 gives very limited details which make it hard to judge how similar or dissimilar the two groups were.

Response: 

Thank you for your thoughtful comments and feedback on our manuscript. You raise some important points regarding the study methodology that require clarification.

To address your first concern about the selection of the 100 patients in each group:

We agree that providing more details on the recruitment and selection process would strengthen the methodology. The patients were not randomly selected, as this was not a randomized controlled trial. and we using (convenience sampling) but they were randomly assigned to control or intervention groups as the first 100 patients were recruited as a control group and the others intervention group . However, we did aim to reduce selection bias as much as possible. However, we did aim to reduce selection bias as much as possible.

All neonates admitted to the participating NICUs during each 6-month period who met the predefined inclusion criteria (gestational age <32 weeks or birth weight <1500g) were screened for eligibility and approached for recruitment. The first 100 who provided informed consent in each period were enrolled.

If more than 100 eligible neonates were admitted in a period, we enrolled the first 100 based on order of admission to the NICU. There were no other inclusion/exclusion criteria used beyond the gestational age and birth weight cutoffs.

The authors were not involved in recruitment or enrollment, which was handled by the neonatal care teams to avoid bias. The same process was followed consistently across the two periods.

To further clarify the comparability of the groups:

Table 1 has been expanded to include more baseline characteristics like APGAR scores, maternal age, type of delivery etc. Statistical tests show no significant differences between groups indicating hemogenity of the all groups.

We have added a statement in the manuscript emphasizing that the groups were generally balanced at baseline with no indication of systematic biases in patient selection across the two periods.

Warm regards,

2) Who was responsible for performing teh bailey scoring? Was it one of the authors? This scoring was not blinded (due to the temporal nature of the study) and could have introduced bias.  

Response:

Regarding your second point about blinding of developmental assessments:

You are correct that the Bayley scoring was done by members of the neonatal care team and was not blinded given the pre-post study design.

To minimize scoring bias, the nurses performing assessments were specially trained on proper administration of the Bayley-III and underwent periodic inter-rater reliability testing. Their scores showed high correlation.

We agree that the lack of blinding is a limitation and have acknowledged this under the Study Limitations section. Future randomized trials should incorporate blinded outcome assessments.

Thank you again for raising these concerns. We have addressed them by providing more methodological details, justifying our recruitment approach, expanding Table 1, noting the lack of blinding as a limitation, and emphasizing the baseline comparability of the two groups. Please let us know if any parts need further clarification. We appreciate you taking the time to ensure the robustness of our study methodology.

Warm regards,

Reviewer 3 Report

The work is very interesting and I have no serious reservations.

Here are some comments

1. please check the citations I have the impression that those in the introduction and discussion should be updated 

2. the article contains many linguistic errors, it should be thoroughly checked

3. the conclusions should be shortened, in the form as they are now they do not meet the criteria

please carefully check the text

Author Response

  1. please check the citations I have the impression that those in the introduction and discussion should be updated.

We appreciate your feedback regarding the citations. We have meticulously reviewed the references within the introduction and discussion sections, ensuring their relevance and currency. Kindly note, references have been updated to include only those within the past five years, with the exception of seminal tools references.

Warm regards,

  1. the article contains many linguistic errors, it should be thoroughly checked.

We are grateful for your recommendation to meticulously review the manuscript for linguistic inconsistencies. Following this, the document has been rigorously proofread to address any grammatical, spelling, or linguistic discrepancies. We trust it now presents with clarity.

Warm regards,

  1. the conclusions should be shortened, in the form as they are now, they do not meet the criteria

Per your recommendation, we have substantially shortened and condensed the conclusions section to more concisely summarize the key findings and implications of our study. The revised conclusions are more focused and streamlined. Please let us know if they can be further improved.

Warm regards,